# Treatment with a VEGFR-2 antibody results in intra-tumor immune modulation and enhances anti-tumor efficacy of PD-L1 blockade in syngeneic murine tumor models

Yanxia Li[1¤a], Nelusha Amaladas[1], Marguerita O'Mahony[1], Jason R. Manro[2], Ivan Inigo[1¤b], Qi Li[1], Erik R. Rasmussen[3], Manisha Brahmachary[3¤c], Thompson N. Doman[2], Gerald Hall[1], Michael Kalos[3¤d], Ruslan Novosiadly[3¤e], Oscar Puig[4☯¤f]*, Bronislaw Pytowski[1☯¤g], David A. Schaer[1☯¤h]

1 Loxo Oncology at Lilly, Eli Lilly and Company, New York City, New York, United States of America, 2 Lilly Research Laboratories, Eli Lilly and Company, Indianapolis, Indiana, United States of America, 3 Lilly Research Laboratories, Eli Lilly and Company, New York City, New York, United States of America, 4 Lilly Oncology, Alexandria Center for Life Sciences, New York City, New York, United States of America

☯ These authors contributed equally to this work.
¤a Current address: Discovery Biology, Regeneron, Tarrytown, New York, United States of America
¤b Current address: Biology, Astra Zeneca, Wilmington, Delaware, United States of America
¤c Current address: Biostatistics, Sanofi, Bridgewater, New Jersey, United States of America
¤d Current address: Next Pillar Consulting, LLC, Philadelphia, Pennsylvania, United States of America
¤e Current address: Translational Medicine, Bristol-Myers Squibb, Princeton, New Jersey, United States of America
¤f Current address: Clinical Biomarkers, BeiGene USA, Inc., Ridgefield Park, New Jersey, United States of America
¤g Current address: Discovery Biology, OncXerna Therapeutics, Inc., Waltham, Massachusetts, United States of America
¤h Current address: Oncology, Pfizer WRD, Pearl River, New York, United States of America
* oscar.puig@beigene.com

**Data Availability Statement:** All relevant data are within the paper and its Supporting information files.

## Abstract

Prolonged activation of vascular endothelial growth factor receptor-2 (VEGFR-2) due to mis-regulation of the VEGF pathway induces aberrant blood vessel expansion, which supports growth and survival of solid tumors. Therapeutic interventions that inhibit the VEGFR-2 pathway have therefore become a mainstay of cancer treatment. Non-clinical studies have recently revealed that blockade of angiogenesis can modulate the tumor microenvironment and enhance the efficacy of concurrent immune therapies. Ramucirumab is an FDA-approved anti-angiogenic antibody that inhibits VEGFR-2 and is currently being evaluated in clinical studies in combination with anti-programmed cell death (PD-1) axis checkpoint inhibitors (pembrolizumab, durvalumab, or sintilimab) across several cancer types. The purpose of this study is to establish a mechanistic basis for the enhanced activity observed in the combined blockade of VEGFR-2 and PD-1-axis pathways. Pre-clinical studies were conducted in murine tumor models known to be responsive to anti-PD-1 axis therapy, using monoclonal antibodies that block mouse VEGFR-2 and programmed death-ligand 1 (PD-L1). Combination therapy resulted in enhanced anti-tumor activity compared to anti-PD-L1 monotherapy. VEGFR-2 blockade at early timepoints post-anti-PD-L1 therapy resulted in a

**Funding:** This work was sponsored by Eli Lilly and Company, but the authors received no specific funding for this work.

**Competing interests:** Y.L. employee and shareholder of Eli Lilly at the time of this work and is currently an employee of Regeneron. N.A. employee and shareholder of Eli Lilly. M.O.M. employee and shareholder of Eli Lilly. J.R.M. employee and shareholder of Eli Lilly. I.I. employee and shareholder of Eli Lilly at the time of this work and is currently an employee of Astra Zeneca. Q.L. employee and shareholder of Eli Lilly. E.R.R. employee and shareholder of Eli Lilly. M.B. employee and shareholder of Eli Lilly at the time of this work and is currently an employee of Sanofi, US. T.N.D. employee and shareholder of Eli Lilly. G. H. employee and shareholder of Eli Lilly. M.K. employee and shareholder of Eli Lilly at the time of this work, and currently Managing Director of Next Pillar Consulting, LLC. M.K. reports stock ownership as a result of employment or advisory roles in: ArsenalBio, Immunai, Cue Biopharma, Nanocell therapeutics, IMV inc., SentiBio, AdicetBio, Orange Grove Bio. Issued patents in the field of cell therapy, licensed by the University of Pennsylvania to Novartis corporation, resulting in royalty distributions. R.N. employee and shareholder of Eli Lilly at the time of this work and is currently an employee of Bristol-Myers Squibb. O.P. employee and shareholder of Eli Lilly. B.P. employee and shareholder of Eli Lilly at the time of this work and is currently an employee of OncXerna Therapeutics, Inc. B.P. reports support for attending meetings and/or travel and stock/ stock options from OncXerna Therapeutics, Inc. D. A.S. employee and shareholder of Eli Lilly at the time of this work and is currently an employee of Pfizer. D.A.S. reports support for attending meetings and/or travel from Pfizer Inc (employee of Pfizer). This does not alter our adherence to PLOS ONE policies on sharing data and materials.

dose-dependent and transient enhanced infiltration of T cells, and establishment of immunological memory. VEGFR-2 blockade at later timepoints resulted in enhancement of anti-PD-L1-driven immune cell infiltration. VEGFR-2 and PD-L1 monotherapies induced both unique and overlapping patterns of immune gene expression, and combination therapy resulted in an enhanced immune activation signature. Collectively, these results provide new and actionable insights into the mechanisms by which concurrent VEGFR-2 and PD-L1 antibody therapy leads to enhanced anti-tumor efficacy.

# Introduction

Angiogenesis, the formation of new blood vessels from pre-existing ones, is an important step in cancer progression [1]. Tumor-induced angiogenesis is highly dysregulated compared to normal angiogenesis, and results in structurally aberrant vessels with insufficient pericyte coverage and leaky nascent vessels [2]. The interplay between aberrant tumor vessels and protumoral immune cells generates a vicious cycle that severely disturbs anti-cancer responses and promotes tumor progression; abnormal tumor vessels promote immune cell evasion, which in turn enhances tumor angiogenesis. This aberrant crosstalk hinders T-cell infiltration into the tumor and impairs T-cell effector functions [3–6].

Overexpression of the soluble pro-angiogenic signal, vascular endothelial growth factor A (VEGF-A), and subsequent engagement of its primary receptor vascular endothelial growth factor receptor-2 (VEGFR-2), are considered key events that drive abnormal angiogenesis [7, 8]. As a result, several therapeutics designed to inhibit the VEGF/VEGFR-2 signaling axis have been developed and introduced into clinical practice [9]. Anti-angiogenic therapies also act to reduce blood supply around tumor cells, depriving cancer cells of oxygen and nutrients, and reducing metastasis to distant sites. Bevacizumab, a monoclonal antibody that binds to and blocks the function of VEGF, was the first anti-angiogenic drug approved by the United States Food and Drug Administration (FDA) [10]. A number of additional anti-VEGF-pathway agents followed, including ramucirumab, a monoclonal antibody that targets VEGFR-2, axitinib, regorafenib, sorafenib, sunitinib, vandetanib, and other small-molecule tyrosine kinase inhibitors that block the activity of VEGF receptors, and ziv-aflibercept, a recombinant fusion protein that sequesters multiple ligands in the VEGF family [11, 12]. Despite demonstrable clinical activity, the overall benefit of anti-angiogenic monotherapies is modest, with moderate improvement of overall survival in some indications, and delayed disease progression without extending overall survival in others [9]. As a result, there is growing interest in combining anti-angiogenic drugs with other treatments. Anti-angiogenic therapies combined with cytotoxic chemotherapy have proven to be a more effective strategy compared to chemotherapy alone [13]. Suppressing the VEGF signaling pathway improves tumor vessel structure and function, resulting in better circulation and lower tumor interstitial pressure. As a result, functional normalization of the vascular system leads to faster and more complete delivery of chemotherapeutics [14].

The role of VEGF/VEGFR-2 in the anti-tumor immune response is another area of increased interest for combined angiogenic therapy approaches. VEGF levels were found to be inversely correlated with tumor-infiltrating leukocytes in human tumors [15], whereas VEGF pathway inhibitors increased leukocyte infiltration into tumors [16, 17]. More recent evidence suggests that this suppressive effect on the anti-tumor immune response is mediated by the VEGF/VEGFR-2 axis. Inhibition of this pathway facilitates T cell migration into the tumor,

and reduces the direct immune inhibitory activity promoted by tumor endothelial cells [18, 19]. Combining anti-angiogenic antibodies with immune checkpoint inhibitors (ICIs) has shown promising results in solid tumors [20–22]. Recent work has shown increases in T cell infiltration following programmed death-ligand 1 (PD-L1) and VEGF blockade combination therapy in patients with hepatocellular carcinoma (HCC) and suggests that targeting VEGF potentially synergizes mechanistically with PD-1/PD-L1 blockade. Analysis of responses showed improved survival in patients with higher inhibitory myeloid signatures receiving VEGF and PD-L1 inhibition, suggesting the combination relieves immune suppression in the tumor microenvironment [23]. However, the mechanisms underlying immunomodulatory activity of agents that target the VEGF/VEGFR-2 axis, and how inhibition of VEGF signaling leads to restoration of productive immunity, have yet to be fully elucidated.

Previous studies using murine tumor models have combined ICIs with anti-angiogenic therapies [4, 24, 25]. Yasuda and colleagues demonstrated that simultaneous blockade of PD-1 and VEGFR-2 produced a synergistic anti-tumor effect in the syngeneic CT26 colon adenocarcinoma model [26]. Allen and colleagues observed that DC101, a mouse surrogate of ramucirumab, stimulated the formation of high endothelial venules within tumors that facilitated enhanced T cell infiltrate, explaining synergistic effects of anti-angiogenic therapy combined with checkpoint inhibition [27]. Similarly, Shigeta and colleagues reported that dual anti-PD-1 and DC101 therapy overcame treatment resistance to either treatment alone, and increased overall survival in both anti-PD-1 therapy-resistant and anti-PD-1 therapy-responsive HCC models [28]. Notably, in immune replete syngeneic mouse models, it has been reported that blockade of VEGFR-2 but not VEGFR-1 or VEGFR-3, regulates intra-tumor immune cell infiltration, but the mechanics of their different roles in immunotherapy combinations require further investigation [12, 29, 30].

In this report, we investigate the efficacy and mechanism of action of combined VEGFR-2 and PD-L1 blockade using the syngeneic models of colorectal (MC38) and breast mammary carcinoma (EMT6) tumors. Combination therapy led to increases in myeloid and T cell infiltration, enhanced innate immune response (e.g., dendritic cell maturation, antigen presentation) and T cell activation, and an immune activation gene expression signature in the tumor. Mice achieving complete tumor regressions after the combination treatment resisted tumor re-challenge, demonstrating the development of immunological memory. Taken together, these results highlight the potential for anti-VEGFR-2 antibodies to relieve innate immune suppression induced by VEGF, providing new insights into the mechanisms by which combined VEGFR-2 and PD-L1 antibody therapy leads to enhanced anti-tumor efficacy.

## Material and methods

### Mice

Female BALB/c and C57BL/6 mice (6–8 weeks of age) were purchased from Harlan Laboratories/Envigo. Mice were anesthetized by vaporizer with isoflurane in a closed chamber, and euthanized by $CO_2$, followed by cervical dislocation to alleviate suffering. Experimental protocols were approved by the Institutional Animal Care and Use Committee and all animal experimental procedures were conducted in accordance with the guidelines of the National Institutes of Health Guide for Care and Use of Animals.

### In vivo tumor studies

In vivo tumor studies and assessment of anti-tumor efficacy have been described in detail previously [30]. Briefly, female BALB/c or C57BL/6 mice (6–8 weeks of age) were subcutaneously implanted in the flank with $1x10^6$ MC38 or $5x10^5$ EMT6-LM2 tumor cells, respectively, on day

0. Both cell lines are PD-L1 positive [31–33]. Rat monoclonal antibodies DC101, MF1, and mF4-31C1 were produced by Eli Lilly and Company as previously described [12, 28, 29]. Rat anti-mouse anti-PD-L1 was generated from a rat of Lou/WS1 strain immunized with recombinant mouse PD-L1-Fc protein. 178G7 was identified based on its binding to PD-L1 (EC50 0.1 nM) and blocking activities against PD-L1 interactions with PD-1 and CD80 (IC50 1.5 nM and 2.5 nM, respectively).

Body weight and tumor volume (TV) were recorded twice a week. TV was calculated as TV $(mm^3) = \pi/6 * length * width^2$. Animals were sacrificed due to progressive disease if tumor burden was greater than 2,500mm$^3$, or growth would surpass 2,500 mm$^3$ before the next measurement. For experiments with secondary tumor re-challenge, mice were re-challenged with 1x10$^6$ MC38 or 5x10$^5$ EMT6-LM2 tumors on the opposite flank of the original tumor injection site. Secondary challenge tumor growth was followed for up to 22 days.

## Immunohistochemistry & immunofluorescence

Immunohistochemistry (IHC) and immunofluorescence (IF) were performed on 5 μm whole tumor sections from the formalin fixed, paraffin embedded tissue blocks. For IF, tumor sections were baked at 60°C for 1 hour followed by deparaffinizing and hydrating using graded concentrations of ethanol to deionized water. Antigen retrieval was performed using DIVA decloaker in a digital electric pressure cooker (decloaking chamber, Biocare Medical) for 10 minutes at 110°C for antigen CD3, ICAM-1, CD31, and Meca-79; or by incubating with proteinase K (Agilent Technologies) for 5 minutes at room temperature for antigen NG2. Following antigen retrieval, all sections were washed and blocked with protein block (Agilent Technologies) for 30 minutes at room temperature. Tissue sections were then incubated for 90 minutes with primary antibody pairs diluted in antibody diluent (Agilent Technologies): CD31, CD3, ICAM1 (Abcam), NG2 (Millipore), or Meca-79 (Novus Biologicals), with appropriate isotype control IgG antibodies, followed by a 60-minute incubation with appropriate secondary antibody pairs in antibody diluent at room temperature. Sections were then counterstained with Hoechst stain (Invitrogen) and mounted under coverslips. Representative fluorescent images were captured using the A1 confocal microscope (Nikon).

IHC was used for quantitative CD3+ analysis. 5 μm whole tumor FFPE sections were baked, deparaffinized and antigens were retrieved in the same manner as IF staining protocol described above. Diaminobenzidine staining was carried out to detect CD3 positive cells and hematoxylin was used as a nuclear counterstain. Analysis was performed in whole slide sections scanned using the Aperio system. Spectrum analysis software was used to determine the percentage of CD3+ cells, and one-way ANOVA was used to determine statistical significance.

## Flow cytometry analysis

After tumors were harvested from individual mice, 2 cubic millimeters of tumor were separated from the whole tumor and were frozen in liquid nitrogen for gene expression analysis. The rest of the tumor tissue was processed into single cells by homogenizing through 40 μm nylon mesh strainers into complete media (RPMI+10% FBS). First, single cell samples were blocked with Fc block (Tonbo), and then incubated with fluorescently labeled antibodies to identify immune markers using murine specific antibodies against CD3, CD4, CD8, CD11b, CD19, CD45, CD11c, CD137, CD25, GITR, Lag3, Tim3, PD1, PDL1, MHCII, and CD86 (eBioscience, Thermo Fisher), and a fixable viability dye (Affymetrix, Thermo Fisher). Appropriate unstained and "Fluorescence Minus One" controls were used. Next, the cells were fixed, permeabilized, and stained with intracellular markers Ki67 and FoxP3 (eBioscience, Thermo Fisher). Labeled single cell suspensions of tissues were collected individually from each mouse

and subjected to flow cytometry following a described gating strategy [34]. Samples were collected on a 4-laser Fortessa X-20 cytometer (BD Biosciences) and analyzed with FlowJo V10 software (Flowjo, LLC, Ashland, OR). The level of infiltration was determined using prism software.

## nCounter gene expression analysis

Total RNA was isolated as described in Schaer et al. [34]. In the nCounter® assay (NanoString® Technologies, Seattle, WA), 50 to 100 ng of RNA was used using mouse-specific nCounter® PanCancer Immune Profiling and PanCancer Pathways Panels. Samples were prepared using an nCounter® Prep Station and codeset/RNA complexes were immobilized on nCounter® cartridges for data collection; data were collected on a NanoString Digital Analyzer.

## Statistical analysis methods

**In vivo tumor study analysis and anti-tumor efficacy measurement.** Tumor volumes were transformed to a log10 scale to equalize the variance across time and treatment. Log10 volume and body weight were separately analyzed using a two-way repeated measures analysis of variance model (RM ANOVA) consisting of time, treatment, and the interaction between time and treatment using the MIXED procedure of the SAS software package (Version 9.4). Spatial Power covariance structure was used to model the correlation of observations across time for the same subject. Kenward-Roger denominator degrees of freedom calculations were used for tests of fixed effects. Post-hoc pairwise t-tests were used to compare tumor volumes and body weights of treated groups to the control group on the summarized day, p-values $\leq 0.05$ were considered statistically significant. The MIXED procedure was also used separately for each treatment group to calculate least squares means and standard errors for each time point for the purpose of plotting and inclusion in summary tables.

**Anti-tumor efficacy definitions.** Efficacy was calculated at the end of the treatment if at least half of the initial number of subjects in the control group remained in the study. Otherwise, efficacy was calculated on the most recent observation day prior to the end of treatment where these conditions were met.

**% Delta T/C.** % *Delta T/C* was defined as 100 times the ratio of the tumor volume change from *Baseline* at time *t* of the treated group versus the tumor volume change from *Baseline* of the control group at time *t*, where *t* is greater than $t_{Baseline}$ and the treated group change from *Baseline* is greater than zero. Baseline tumor volume is the grand mean of all tumors at $t_{Baseline}$.

**% Tumor regression.** % *Regression* was defined as 100 times the ratio of the tumor volume change from *Baseline* of the treated group versus *Baseline* tumor volume at time *t*, where *t* is greater than $t_{Baseline}$ and the treated group change from *Baseline* is less than or equal to zero.

**% Tumor Growth Inhibition (TGI).** % *TGI* was defined as 100 minus % *Delta T/C* or 100 minus % *Regression* as applicable.

**Flow cytometry analysis.** Data from all flow cytometry experiments were analyzed using either two-sample t-test, one-way, or two-way ANOVA models as appropriate for the design of each experiment. Dunnett's method was used for ANOVA analyses to compare treated groups to control. A type I error rate of 5% (p-value $\leq 0.05$) was used to denote statistical significance. Statistical analyses were performed using either JMP (ver. 15.2.0, SAS Institute Inc., Cary, NC) or GraphPad Prism (ver. 7) software packages. Results were summarized and plotted as means ± standard error.

**nCounter gene expression analysis.** nCounter RNA count data were normalized using the geometric mean of the housekeeping genes. The log$_2$ transformed normalized counts were

first analyzed to detect and exclude statistical outliers. Observations were considered statistical outliers if the absolute value of its studentized residual from a one-way ANOVA model was greater than 3.5. The $log_2$ transformed data were then analyzed using one-way ANOVA by gene followed by post-hoc t-tests to compare treated groups to control. The significance of the combination treatment effect for gene expression was also tested using an ANOVA contrast statement equivalent to comparing the observed combination effect to its expected if additive effect as defined by the Bliss Independence method. The treatment fold change from control was estimated by taking the anti-log of the $log_2$ mean differences.

The results from the statistical analysis were then used in biological pathway analyses using Ingenuity Pathway Analysis. Data visualizations were done using Spotfire® software (TIBCO, Somerville, MA).

## Results

### Anti-VEGFR-2 treatment increases intra-tumor T cell infiltration in a dose-dependent manner and reduces vessel density in MC38 mouse tumor model

It has been reported that optimal inhibition of angiogenesis leads to a more homogenous distribution of functional tumor vessels, which facilitates intra-tumor immune cell infiltration [35, 36]. Since multiple VEGF receptors could have an effect on vessel development, we first investigated and confirmed if VEGFR-2 has a major role in T cell infiltration. MC38 bearing mice were treated with a single dose of 40 mg/kg DC101 (anti-VEGFR-2), MF1 (anti-VEGFR-1), or mF4-31C1 (anti-VEGFR-3). After 15 days of therapy, only treatment with DC101, the anti-VEGFR-2 antibody, significantly increased the frequency of CD3+ T cells at the evaluated timepoint, compared to the control treatment (Rat IgG) (S1 Fig).

Having confirmed the potential role of VEGFR-2 in controlling T cell infiltration, we next evaluated whether the DC101-dependent intra-tumor T cell infiltration was dose dependent and how it related to anti-tumor efficacy. DC101 (5 mg/kg, 20 mg/kg, and 40 mg/kg) was administered to MC38 tumor-bearing mice, and T cell infiltration into tumors was measured by IHC and flow cytometry. At day 21, the tumor volumes for the lowest dose of DC101 (5 mg/kg) were not statistically different from control, whereas the mid (20 mg/kg) and high (40 mg/kg) doses of DC101 both showed dose-dependent statistically significant tumor growth inhibition (% *Delta T/C* = 46.7% and 35.4%, respectively) (Fig 1A). After 7 days of therapy, tumors were collected and analyzed by immunofluorescence and flow cytometry to check for changes in angiogenic markers and T cell frequency. Consistent with the tumor growth inhibition data, tumor vessel density based on CD31 and pericyte marker NG2 staining did not change in the 5 mg/kg group compared to the control IgG treatment group (Fig 1B). In contrast, both the 20 mg/kg and 40 mg/kg doses of DC101 treatments reduced vessel density (Fig 1B). By IHC, T cell infiltration appeared to be significantly increased only at the highest dose of DC101 treatment compared to control at day 15 (Fig 1C). Comparatively, a similar result was observed using flow cytometry, with a trend toward a dose-dependent increase in T cell frequency compared to control at day 15 (Fig 1D). At day 19, increased T cell infiltration was only maintained in the 40 mg/kg of DC101 treatment cohort.

High endothelial venules (HEV) have been found in human solid tumors and their presence correlates with lymphocyte infiltration [27]. To investigate further the mechanism of increased T cell infiltration after DC101 therapy, we evaluated if DC101 treatment induces HEV formation in the same MC38 tumor model. Immunohistochemical analysis of the tumor samples revealed the presence of more HEV, as detected by Meca 79 staining, in tumors from the 40mg/kg DC101 treated mice, compared to tumors from control treated mice (Fig 1E).

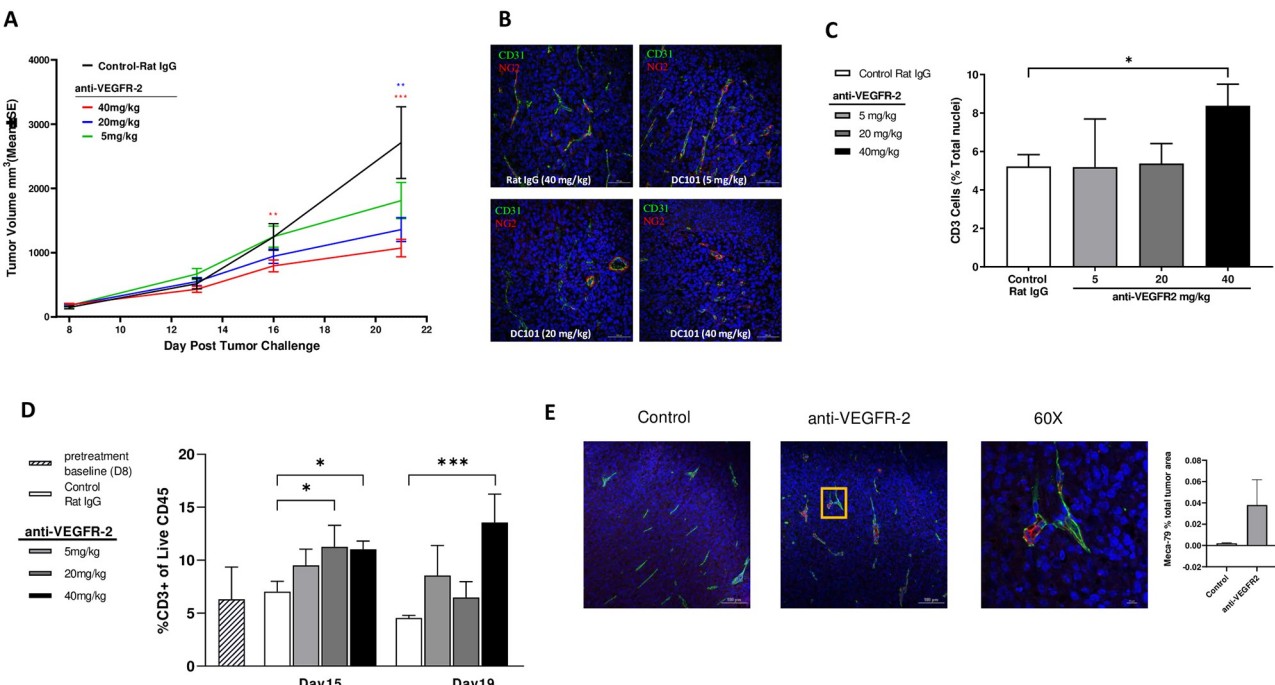

**Fig 1. VEGFR-2 blockade induces anti-tumor effects and increases T cell infiltration in tumors.** (**A**) Mean MC38 tumor volumes in BALB/c mice during 5 mg/kg (green), 20 mg/kg (blue), and 40 mg/kg (red) DC101 treatment (measured twice per week for 3 weeks). (**B**) Representative IHC images of CD31 (green), α-smooth muscle actin NG2 (red), and cell nuclei (Hoechst) immunostaining on tumor sections from day 15 (scale bars: 100 μM). (**C**) Percentage of CD3+ cell infiltrates into tumor by IHC on day 15 by IHC (n = 5 for each group). (**D**) Percentage of CD3+ cells in tumor measured by flow cytometry analysis on day 15 and day 19 (n = 5 for each group). (**E**) Representative IHC images of CD31 (green), Mega 79 (red), and cell nuclei (Hoechst) immunostaining on tumor sections from day 13. Right bottom graph represents the 60x amplification (scale bar: 10 μM) of inset in left bottom graph (scale bar: 100 μM). Bar graph showed the quantification of Meca 79+ cell area in total tumor area among the treatment groups (n = 4 to 5 for each group). Statistical significance: *P $\leq$ 0.05, **P $\leq$ 0.01, ***P $\leq$ 0.001. Error bars indicate SEM.

## Combination of anti-VEGFR-2 with anti-PD-L1 antibodies improves the antitumor efficacy, and induces immune memory in murine tumor models

To better understand the combination potential of VEGFR-2 with PD-L1 therapy and determine if the increased intra-tumor T cell infiltration induced by anti-VEGFR-2 therapy is a primary mechanism improving the benefit of PD-L1 checkpoint blockade, we evaluated the combined therapy in MC38 and an additional syngeneic tumor model EMT-6. PD-L1 therapy has been well characterized in both models allowing us to confirm combination effect and extrapolate the potential contribution of VEGFR-2. In the MC38 tumor-bearing mice treated with anti-VEGFR-2 or anti PD-L1 antibodies as single agents starting on day 3 (Fig 2A), a slight delay in tumor growth was observed (Fig 2B and 2C). Concurrent administration of anti-VEGFR-2 with anti-PD-L1 antibodies achieved greater tumor growth delay, with two complete responders (Fig 2D). In the EMT6-LM2 tumor model, moderate tumor growth inhibition was observed with anti-PD-L1 and anti-VEGFR-2 antibody monotherapies (anti-PD-L1, complete response [CR] = 1; anti-VEGFR-2, CR = 2) (Fig 2E and 2G). Stronger tumor growth inhibition was observed when anti-PD-L1 and anti-VEGFR-2 therapies were combined, with seven complete responders (Fig 2H).

Having established that these models showed primary combinational benefit, we then tested if this extended to the development of immunological memory in complete responders. All complete responders in the representative experiment from Fig 2 were re-challenged with either MC38 (Fig 2I) or EMT6-LM2 (Fig 2J) tumor cells on the opposite flank of the original

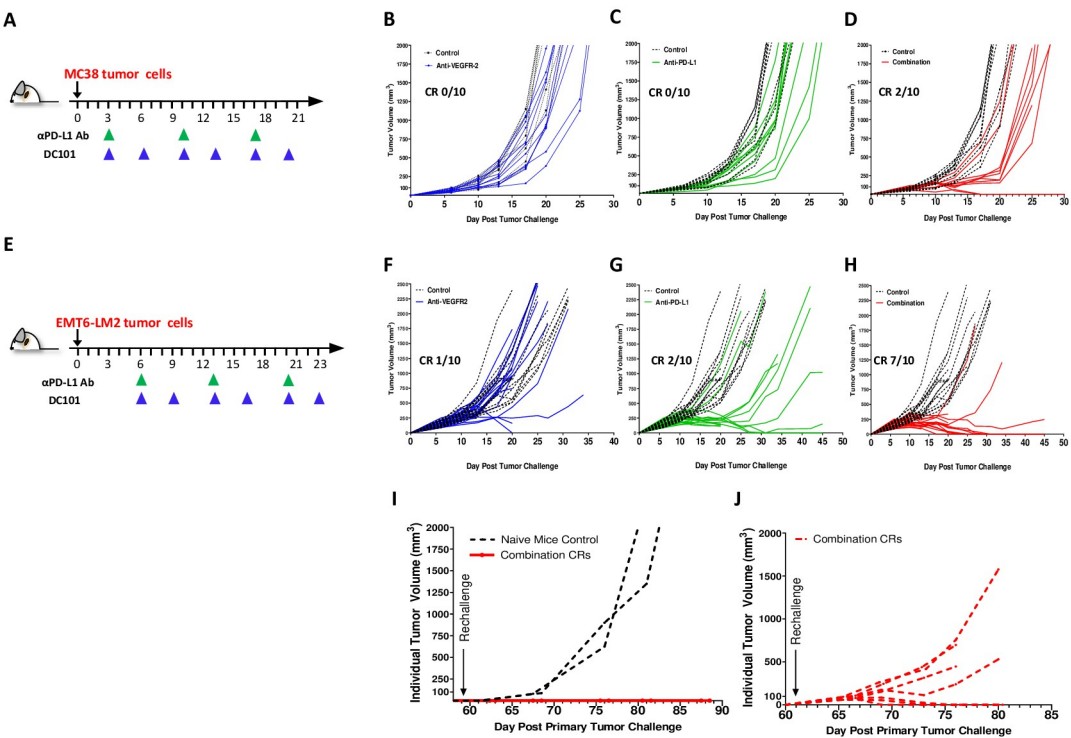

**Fig 2. Combination of VEGFR-2 blockade with anti-PD-L1 antibodies improves antitumor efficacy and induces immunological memory in MC38 and EMT6-LM2 syngeneic murine model. (A)** Schema of MC38 tumor-bearing mice treatment regimen. Dosing started at day 3. Subsequently, DC101 and Rat IgG were given i.p. 40 mg/kg, twice per week for 3 weeks. Anti-PD-L1 (178G7) was given 500 μg/mouse, i.p., once per week, 3 times in total. **(B)** Individual tumor growth curves of the DC101 treatment group are superimposed to Rat IgG control group. **(C)** Individual tumor growth curves of the anti-PD-L1 treatment group are superimposed to Rat IgG control group. **(D)** Individual tumor growth curves of the combination treatment group (DC101 with anti-PD-L1) are superimposed to Rat IgG control group. **(E)** Schema of EMT6-LM2 tumor-bearing mice treatment regimen. **(F)** Individual tumor growth curves for DC101. Dosing started at day 6. Subsequently, DC101 and Rat IgG were given i.p. 40 mg/kg, twice per week for 3 weeks. Anti-PD-L1 (178G7) was given 500 μg/mouse, i.p., once per week, 3 times in total. **(G)** Individual tumor growth curves for anti-PD-L1 (178G7). **(H)** Individual tumor growth curves for the combination treatment group. **(I)** Individual tumor growth curves for the 2 CR mice from **(D)** (red lines) and 2 naïve mice (black lines), which were re-challenged with MC38 tumor cells on the opposite flank on day 60. **(J)** Individual tumor growth curves for 7 CR mice from **(H)** (red lines), which were re-challenged with EMT6-LM2 tumor cells on the opposite flank on day 60. CR, complete response. *P ≤0.05, **P ≤0.01, ***P ≤0.001. Error bars indicate SEM.

tumor on day 60, more than 30 days after cessation of treatment. All mice that achieved CRs after the primary tumor challenge in the MC38 model, rejected secondary re-challenged tumors, implying the formation of a memory T cell response during the first tumor rejection. This response was not observed in naïve mice implanted with MC38 tumors as a control (Fig 2I). In the EMT6-LM2 model, three of seven CRs resisted re-challenge after the primary tumor challenge (Fig 2J), again implying the development of immunological memory. However, in four of seven CRs, it was observed the re-challenged EMT6-LM2 tumors were able to grow, albeit at a slightly slower rate (Fig 2J compared to Fig 2H), suggesting that the development of immune memory is only partial.

## Anti-VEGFR-2 immune effects synergizes with anti-PD-L1 to enhance immunomodulation

To understand the combination effects between anti-VEGFR-2 and anti-PD-L1 therapy at the molecular level, changes in immune cell subset frequency were examined in MC38 tumors

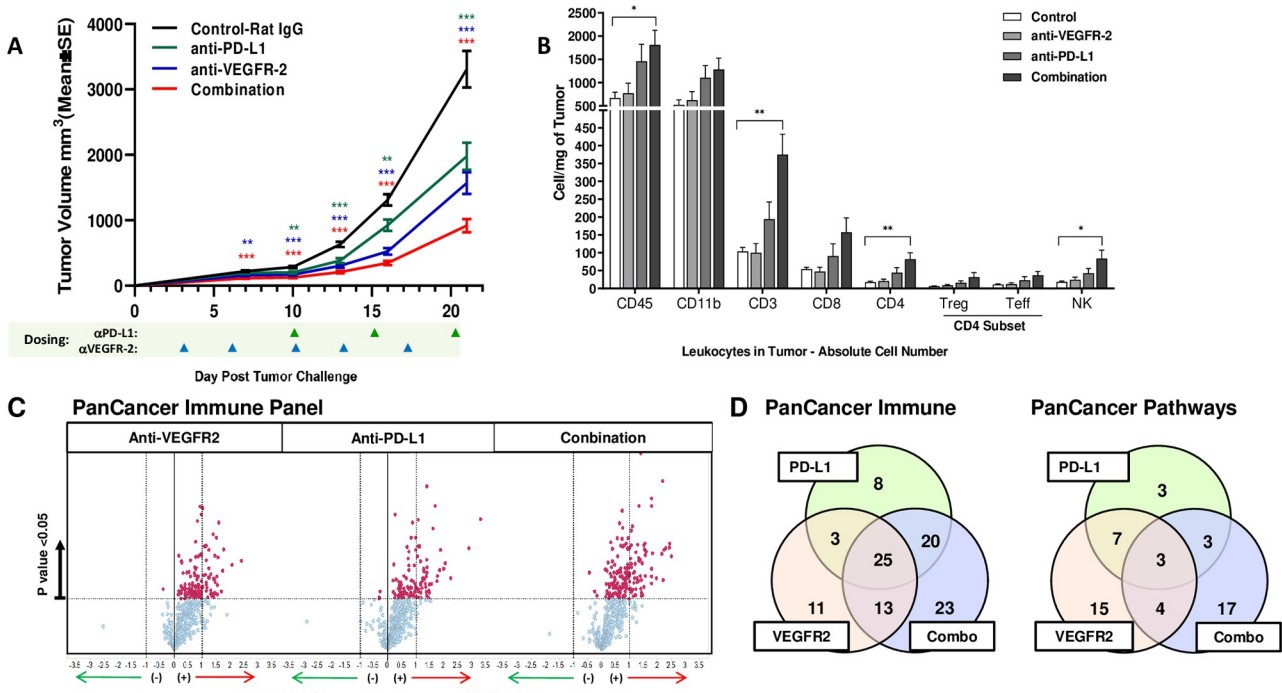

**Fig 3. Combination of VEGFR-2 blockade with anti-PD-L1Ab results in antitumor effects, increased T cell infiltration, and unique intra-tumor gene upregulation signature in MC38 syngeneic murine model. (A)** Mean MC38 tumor volumes in C57B16 mice for mechanism of action study, according to treatment. $^*$P $\leq$0.05, $^{**}$P $\leq$0.01, $^{***}$P $\leq$0.001. Error bars indicate SEM. **(B)** Absolute leukocyte subset numbers in tumor collected on day 20. $^*$P $\leq$0.05, $^{**}$P $\leq$0.01 **(C)** nCounter analysis using PanCancer Immune Profiling in MC38 tumors at day 20; volcano plots show log2-fold change of gene expression in the treatment groups, compared with the control group. Differentially expressed genes (DEGs) that display greater or less than 2-fold differences are indicated by red and green arrows, respectively. Red data points indicate p<0.05 (one-way ANOVA). **(D)** Venn diagrams show the number of shared and treatment-specific DEGs across experimental groups from PanCancer Immune Profiling (left) and PanCancer Pathways panel (right).

from a separate mechanism of action (MOA) study. Tumor volume changes for this independent study are shown in Fig 3A. Anti-PD-L1 monotherapy increased intra-tumor immune cell infiltration of all cell subsets analyzed, with statistically significant changes in CD3+ T cell, CD4+ T cell, and natural killer (NK) cells. The combination of anti-VEGFR-2 with anti-PD-L1 antibodies further enhanced infiltration of the same immune cell subsets (Fig 3B).

To further understand the mechanisms driving the observed combinational effects, MC38 tumor samples collected from treated animals were evaluated by high-content gene expression analysis, using the murine-specific nCounter PanCancer Immune Profiling and PanCancer Pathways panels (Fig 3C). Direct analysis of differentially expressed genes in the experimental treatment groups compared to control samples revealed that anti-VEGFR-2 and anti-PD-L1 monotherapies induced both shared and non-overlapping changes in expression of immune-related genes (Fig 3D). Furthermore, anti-VEGFR-2 and anti-PD-L1 combination had the greatest impact on gene expression profiles within tumors, with biological pathways modulated in both the monotherapy groups more strongly modulated in the combination group (Fig 3C and Table 1). When compared to monotherapies, pathway analysis in the combination group revealed enrichment of pathways generally associated with T helper cell signaling, T cell activation, dendritic cell (DC)/NK cells and cytokine signaling, as well as pathways associated

**Table 1. Unique intra-tumor gene upregulation signature in MC38 syngeneic murine model.** Ingenuity Pathway Analysis (IPA) using both the **(A)** PanCancer Immune Profiling panel and **(B)** PanCancer Pathways panel lists DEGs with fold-change +/-1.5 and p<0.05. Pathway significantly enriched if log (B-H p-value) ≥1.3.

(A) Nanostring PanCancer Immune panel:

| Functional Class | Ingenuity Canonical Pathways | -LOG(B-H p-value) | | |
|---|---|---|---|---|
| | | PD-L1 (178G7) | VEGFR2 (DC101) | Combo (PD-L1 +VEGFR2) |
| T Helper Cells Signaling/ Differentiation | iCOS-iCOSL Signaling | 12.2 | 10.9 | 18.9 |
| | CD28 Signaling | 12.2 | 10.6 | 18.4 |
| | Differentiation | 10.8 | 11.8 | 21.6 |
| T-cell Activation/Signaling | Role of NFAT in Regulation of the Immune Response | 10.6 | 10.1 | 18.3 |
| | PKCθ Signaling | 9.65 | 6.82 | 14.2 |
| | TCR Signaling | 12.6 | 12.6 | 12.6 |
| Cytokine Mediated Inflammation | IL-4 Signaling | 6.02 | 4.53 | 8.72 |
| | Role of JAK1 & JAK3 in γc Cytokine Signalling | 3.92 | 3.82 | 9.42 |
| | Regulation of IL-2 Expression in Act.&Anergic T Cells | 3.72 | 2.55 | 7.67 |
| Pro-Inflammatory Activity | TREM1 Signaling | 3.93 | 6.26 | 9.47 |
| | IL-12 Signaling and Production in Macrophages | 3.76 | *n.s.* | 6.98 |
| Pro-Apoptotic Activity | Nur77 Signaling in T Lymphocytes | 7.42 | 5.7 | 12.3 |
| | Calcium-induced T Lymphocyte Apoptosis | 7.07 | 4.12 | 10.2 |
| | CTLA4 Signaling in Cytotoxic T Lymphocytes | 4.53 | 2.33 | 9.57 |
| Dendritic Cells | Dendritic Cell Maturation | 9.65 | 11.5 | 19.9 |
| DC/NK Crosstalk | Crosstalk between Dendritic Cells and Natural Killer Cells | 12.2 | 13 | 19.9 |
| Natural Killer Cell Signaling | Natural Killer Cell Signaling | 3.14 | 1.34 | 5.52 |
| B Cell Development and Signaling | B Cell Development | 10.9 | 13.9 | 15.5 |
| | B Cell Receptor Signaling | 3.04 | 4.6 | 3.93 |

(B) Nanostring PanCancer Pathways panel:

| Functional Class | Ingenuity Canonical Pathways | -LOG(B-H p-value) | | |
|---|---|---|---|---|
| | | PD-L1 (178G7) | VEGFR2 (DC101) | Combo (PD-L1 +VEGFR2) |
| T Helper Cells Signaling/ Differentiation | Th2 Pathway | 5.65 | 4.98 | 7.63 |
| | Th1 and Th2 Activation Pathway | 6.15 | 4.53 | 7.04 |
| | Th1 Pathway | 5.83 | 4.12 | 5.47 |
| | IL-12 Signaling and Production in Macrophages | 6.81 | 7.31 | 6.74 |
| Dendritic Cells/Natural Killer Cells | Crosstalk between Dendritic Cells and Natural Killer Cells | 4.3 | 5.11 | 6.74 |
| | Dendritic Cell Maturation | 5.22 | 4.54 | 4.91 |
| Cytokine Mediated Inflammation | HMGB1 Signaling | 4.61 | 8.41 | 6.74 |
| | NF-B Signaling | 5.16 | 8.47 | 5.96 |
| | IL-6 Signaling | 4.61 | 7.37 | 5.47 |
| | IL-10 Signaling | 2.03 | 4 | 5.43 |
| | PKC-Ө Signaling in T Lymphocytes | 4.61 | 6.26 | 4.39 |
| Cytokine Signaling | Role of JAK1 and JAK3 in γc Cytokine signaling | 6.81 | 6.35 | 6.74 |
| | IL-9 Signaling | 3.6 | 4.49 | 4.73 |
| | Differential Regulation of Cytokine Production in Macrophages and T Helper Cells by IL-17A and IL-17F | 3.04 | 2.85 | 4.51 |
| TLR Signaling | Toll-like Receptor Signaling | 4.29 | 6.35 | 5.37 |

with pro-inflammatory and pro-apoptotic activity. The top scoring pathways were related to T cell and DC/NK cell functions (Table 1). In summary, VEGFR-2 inhibition alone causes alterations in the expression of immune genes related to DC maturation and T cell activation, and this effect was enhanced when combined with anti-PD-L1 therapy.

### Flow cytometry confirms increased T cell activation and antigen presentation in response to anti-VEGFR-2 and anti-PD-L1 combination therapy

To confirm the effects of combination therapy on T cell activation and antigen presentation, EMT6-LM2 tumors from the experiment shown in Fig 2, where the combination effect was more pronounced, were collected 20 days after tumor implantation to examine changes in immune cell activation markers by flow analysis (the experimental design and tumor growth curves are shown in Fig 2E–2H). These analyses confirmed the increased T cell activation phenotype observed molecularly after anti-VEGFR-2 and anti-PD-L1 therapy, as evidenced by a higher frequency of proliferating (ki67+) CD8 cells and increased frequency of CD137+, CD25 + and CD8+ T cells, compared to the control group (Fig 4A). In this model, anti-PD-L1 monotherapy resulted in the strongest effect on T cell activation markers, while combination treatment had a less pronounced effect on T cell activation makers. However, a caveat is that combination therapy was very efficacious in reducing tumor size (Fig 2H), therefore remaining tumor material available for flow analysis was limiting, confounding data interpretation. Both anti-VEGFR-2, anti-PD-L1 monotherapies as well as combination therapy, resulted in an enhanced antigen presentation profile on myeloid subsets, specifically on macrophages and Ly6CG-population, as reflected by the upregulation of MHCII. In comparison to the control group (where approximately 20% of macrophages are MHCII+) anti-VEGFR-2 and anti-PD-L1 monotherapies as well as combination therapy resulted in more MHCII+ macrophages (approximately 40%, 80%, and 90% respectively) (Fig 4B **bottom**). Moreover, combination therapy significantly increased the level of MHCII expression on both macrophages and Ly6CG- population, as shown by MHCII MFI (Fig 4B **top**). Furthermore, compared to the control group, anti-VEGFR-2, anti-PD-L1 monotherapies and combination therapy resulted in more MHCII and CD86 double positive macrophages, specifically, 2.9X, 4.3X, and 4.3X CD86, MHCII and double positive macrophages, respectively (Fig 4C). Neither anti-VEGFR-2 nor anti-PD-L1 monotherapy, nor the combination, significantly affected Myeloid-Derived Suppressor Cells (identified by Ly6G+ and Ly6C+). Finally, anti-VEGFR-2 therapy resulted in upregulation of PD-L1 expression on macrophages (Fig 4D).

## Discussion

In this study, we evaluated the functional and molecular consequences of anti-VEGFR-2 therapy as monotherapy and in combination with anti-PD-L1 checkpoint blockade. Delayed tumor growth induced by inhibition of tumor angiogenesis with DC101 was associated not only with vasculature normalization, but also HEV formation. This suggests that the increased T cell infiltration observed after DC101 therapy resulted not only from inhibition of tumor angiogenesis but also through activation of the vasculature causing T cell recruitment. It is possible that tumor infiltrated T cells are further activated by the treatment to recognize tumor antigens, but our experiments were not designed to address this question and additional research will be needed. Finally, we did not observe any changes in NK cell percentages or counts, suggesting NK cells are not critical driving the observed response.

VEGF-A is a multifunctional factor which plays an important role in angiogenesis. It regulates vascular permeability and promotes vascular endothelial cell proliferation, migration, and survival [37–39]. Using specific antagonist antibodies, we showed that these effects are dependent on VEGFR-2, and not VEGFR-1 or VEGFR-3. VEGF-A induces signaling through the VEGFR-1 and VEGFR-2 receptors, while the related proteins VEGF-C and VEGF-D signal through the VEGFR-3 receptor involved in lymph angiogenesis and tumor angiogenesis. While the three VEGF receptors can have overlapping and unique functions in cancer, our

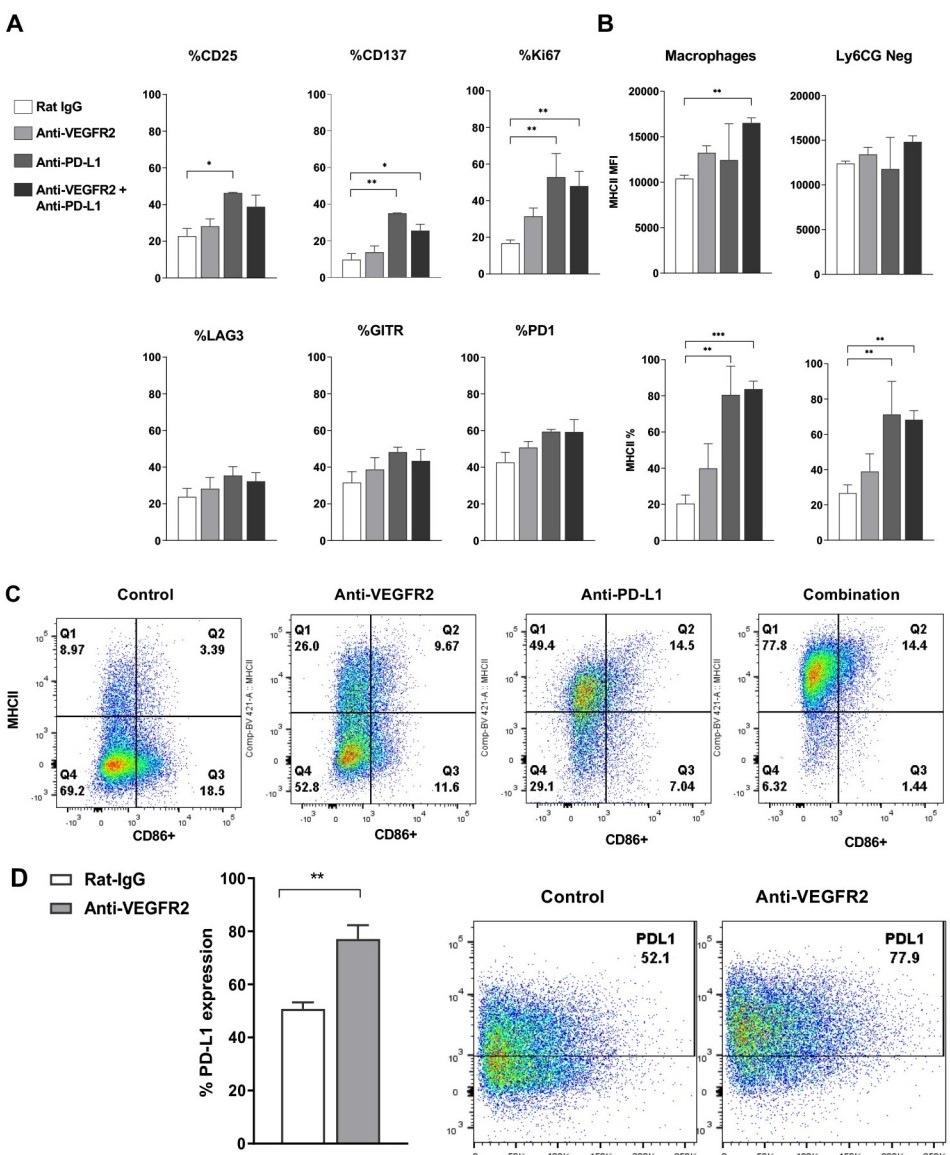

**Fig 4. Increased T cell activation and antigen presentation in response to anti-VEGFR-2 and anti-PD-L1 treatment in murine EMT6-LM2 tumor model by flow cytometry.** EMT6-LM2 tumor samples for each group (n = 5) were collected on day 20 from study shown in Fig 2E. **(A)** The expression of activation markers CD137, CD25, GITR, Ki67, Lag3, and PD1 on intra-tumor CD8 cell were assessed for DC101, anti-PD-L1, or combination treatment. **(B)** Percentage of MHCII+ cells and MHCII expression level on tumor macrophages and Ly6CG Neg population were assessed for DC101, anti-PD-L1, or combination treatment. **(C)** Percentages of double positive expression for CD86 + and MHCII+ on tumor macrophages were analyzed and one set of representative plots is shown. **(D)** PD-L1 expression on tumor infiltrating macrophages was analyzed and one set of representative plots is shown. $^*$P $\leq$0.05, $^{**}$P $\leq$0.01, $^{***}$P $\leq$0.001. Error bars indicate SEM.

data suggest that VEGF-A, through engagement of VEGFR-2 modulates tumor immune infiltration.

TILs levels have also been shown to correlate with the efficacy of ICI therapy, suggesting treatments that enhance T cell infiltration could promote the efficacy of ICIs [40]. This is further supported by previous findings that anti-VEGFR-2 treatment can sensitize tumors to checkpoint blockade by inducing T cell infiltration; however, treatment also leads to PD-L1

expression which can inhibit a T cell response [27, 41]. Likewise, anti-VEGFR-2 therapy led to enhanced T cell infiltration into tumors. However, inhibition additionally led to increased intra-tumor immune inflammation, which led to upregulation of PD-L1 on macrophages as well. This suggests that, while VEGFR-2 inhibition leads to increased immune inflammation, the feedback inhibition of the PD-1/PD-L1 pathway in the tumor limits immune activation. Notably, the combined treatment led to immunological memory, since animals that responded to treatment were resistant to tumor rechallenge.

Several groups have recently reported the combinational effect of angiogenic blockade with ICIs using either PD-1 or PD-L1 in murine tumor models [27, 28, 42]. These studies suggest a similar mechanism of combinatorial efficacy as shown here, further highlighting the potential of anti-angiogenic combination with ICI. The combination appears to target two mechanisms that suppress anti-tumor immunity; blocking the PD-1/L1 axis, in tandem with increased immune infiltration and inflammation induced by VEGF axis inhibition, leads to a fully integrated anti-tumor immune response.

The combination of anti-angiogenic molecules with ICIs holds great promise. Recent Phase 3 trials have shown the benefit of combining PD-1 or PD-L1 antibodies with anti-VEGF reagent, bevacizumab, or anti-VEGFR agent, axitinib, resulting in FDA approval for the combinations across multiple tumor types including in an ICI-experienced setting (NCT02684006, lung; NCT03434379, hepatocellular carcinoma; and NCT02366143, kidney). Several clinical trials combining the anti-VEGFR-2 antibody, ramucirumab, and PD1 or PD-L1 antibodies are ongoing in several indications, including non-small-cell lung cancer, recurrent/metastatic head and neck squamous cell carcinoma, and mesothelioma (NCT03904108, NCT03650764, NCT03502746), and early indications suggest the results of this combination will be promising.

The results presented here show that blocking VEGFR-2 improves T cell infiltration into tumors while PD-L1 antibodies protect these T cells from exhaustion, resulting in enhanced antitumor activity and immunological memory. These data provide further support for the therapeutic strategy of simultaneous VEGFR-2 and PD-L1 inhibition for the treatment of patients with solid tumors.

## Supporting information

**S1 Fig. VEGFR-2 blockade induces tumor T cell infiltration.** MC38 tumor bearing mice were treated with 40mg/kg Rat IgG, MF1 (anti-VEGFR-1), DC101 (anti-VEGFR-2) and mF4-31C1 (VEGFR-3) at a single dose, and then tumors were collected on day 14. (**A**) Representative IHC images of CD31 (green), α-smooth muscle actin NG2 (red) and cell nuclei (Hoechst) immunostaining on tumor sections from day 15; (**B**) Percentage of CD3+ cell infiltrates by IHC (calculated as CD3+ cells over Hoechst positive nuclei) in each tumor sample on day 15 (n = 3 for each group).
(TIF)

## Author Contributions

**Conceptualization:** Yanxia Li, Nelusha Amaladas, Bronislaw Pytowski, David A. Schaer.

**Data curation:** Yanxia Li, Nelusha Amaladas, Marguerita O'Mahony, Jason R. Manro, Ivan Inigo, Qi Li, Erik R. Rasmussen, Manisha Brahmachary, Thompson N. Doman, Gerald Hall, Michael Kalos, Ruslan Novosiadly, Oscar Puig.

**Formal analysis:** Yanxia Li, Nelusha Amaladas, Marguerita O'Mahony, Jason R. Manro, Ivan Inigo, Qi Li, Erik R. Rasmussen, Manisha Brahmachary, Thompson N. Doman, Gerald Hall, Michael Kalos, Ruslan Novosiadly, Oscar Puig.

**Investigation:** Yanxia Li, Nelusha Amaladas, Oscar Puig.

**Methodology:** Yanxia Li, Nelusha Amaladas, Marguerita O'Mahony, Jason R. Manro, Ivan Inigo, Qi Li, Erik R. Rasmussen, Manisha Brahmachary, Thompson N. Doman, Gerald Hall, Michael Kalos, Ruslan Novosiadly, Oscar Puig.

**Supervision:** Oscar Puig, Bronislaw Pytowski, David A. Schaer.

**Writing – original draft:** Yanxia Li, Nelusha Amaladas.

**Writing – review & editing:** Marguerita O'Mahony, Jason R. Manro, Ivan Inigo, Qi Li, Erik R. Rasmussen, Manisha Brahmachary, Thompson N. Doman, Gerald Hall, Michael Kalos, Ruslan Novosiadly, Oscar Puig, Bronislaw Pytowski, David A. Schaer.

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
