## [Decision Letter · Decision Letter 0]

8 Feb 2022

PONE-D-21-39856Treatment with a VEGFR-2 antibody results in intra-tumor immune modulation and enhances anti-tumor efficacy of PD-L1 blockade in syngeneic murine tumor modelsPLOS ONE

Dear Dr. Puig,

Thank you for submitting your manuscript to PLOS ONE. After careful consideration, we feel that it has merit but does not fully meet PLOS ONE’s publication criteria as it currently stands. Therefore, we invite you to submit a revised version of the manuscript that addresses the points raised during the review process by both Reviewers.

We look forward to receiving your revised manuscript.

Kind regards,

Francesco Bertolini, MD, PhD

Academic Editor

PLOS ONE

Journal Requirements:

3. As part of your revision, please complete and submit a copy of the Full ARRIVE 2.0 Guidelines checklist, a document that aims to improve experimental reporting and reproducibility of animal studies for purposes of post-publication data analysis and reproducibility: https://arriveguidelines.org/sites/arrive/files/Author%20Checklist%20-%20Full.pdf (PDF). Please include your completed checklist as a Supporting Information file. Note that if your paper is accepted for publication, this checklist will be published as part of your article.

[Y.L. employee and shareholder of Eli Lilly at the time of this work and is currently an employee of Regeneron.

N.A. employee and shareholder of Eli Lilly.

M.O.M. employee and shareholder of Eli Lilly.

J.R.M. employee and shareholder of Eli Lilly.

I.I. employee and shareholder of Eli Lilly at the time of this work and is currently an employee of Astra Zeneca.

Q.L. employee and shareholder of Eli Lilly.

E.R.R. employee and shareholder of Eli Lilly.

M.B. employee and shareholder of Eli Lilly at the time of this work and is currently an employee of Sanofi, US. 

T.N.D. employee and shareholder of Eli Lilly.

G.H. employee and shareholder of Eli Lilly.

M.K. employee and shareholder of Eli Lilly at the time of this work, and currently Managing Director of Next Pillar Consulting, LLC. M.K. reports stock ownership as a result of employment or advisory roles in: ArsenalBio, Immunai, Cue Biopharma, Nanocell therapeutics, IMV inc., SentiBio, AdicetBio, Orange Grove Bio.  Issued patents in the field of cell therapy, licensed by the University of Pennsylvania to Novartis corporation, resulting in royalty distributions.

R.N. employee and shareholder of Eli Lilly at the time of this work and is currently an employee of Bristol-Myers Squibb.

O.P. employee and shareholder of Eli Lilly.

B.P. employee and shareholder of Eli Lilly at the time of this work and is currently an employee of OncXerna Therapeutics, Inc. B.P. reports support for attending meetings and/or travel and stock/stock options from OncXerna Therapeutics, Inc.

D.A.S. employee and shareholder of Eli Lilly at the time of this work and is currently an employee of Pfizer. D.A.S. reports support for attending meetings and/or travel from Pfizer Inc (employee of Pfizer).] 

Reviewers' comments:

Reviewer's Responses to Questions

**Comments to the Author**

1. Is the manuscript technically sound, and do the data support the conclusions?

Reviewer #1: Yes

Reviewer #2: Partly

2. Has the statistical analysis been performed appropriately and rigorously? 

Reviewer #1: Yes

Reviewer #2: Yes

3. Have the authors made all data underlying the findings in their manuscript fully available?

Reviewer #1: Yes

Reviewer #2: Yes

4. Is the manuscript presented in an intelligible fashion and written in standard English?

Reviewer #1: Yes

Reviewer #2: Yes

5. Review Comments to the Author

Reviewer #1: This manuscript deals with the effect on tumor growth of a combination of anti-VEGFR2 and anti-PDL1 antibodies in murine models.

It is evident that the combination therapy is more effective compared to monotherapy. This effect is accompanied by a stronger T cell infiltration, activation of macrophages and immunological memory to further rechallenge with tumor cells.

Overall, the manuscript is well organized and written and the flow of experiments are straightforward.

I suggest some improvements to further reinforce the main message of this work.

a- the analysis of infiltrating lymphocytes is not well described. Indeed, it is not clear how large are the tumors analyzed, whether a whole scan of tumor sections have been performed, and whether a software of analysis has been used to get the level of infiltration. Also, no analysis of markers involved in lymphocyte migration and localization has been considered. It is conceivable that T cells and myeloid cells can home to the tumor, expressing some receptors involved in endothelial migration.

b- it is not stated that the cell lines used in the tumor murine models are indeed PDL-1 positive, this is important to support the treatment with anti PDL-1 antibodies.

c- all the data are coming from the in vivo murine models. It would be fine to demonstrate that TIL, splenic cells, lymph node T cells, PB lymphocytes  can recognize  the tumor cells in vitro. In other words, are CTLs present in the mice-treated specific for the tumor cells used to induce tumors? Do NK cells play a role in killing tumor cells, and can the antibodies used for therapy trigger ADCC?

Reviewer #2: Li et al. uncovered the treatment with a VEGFR-2 antibody to be potentially associated with intra-tumor immune modulation and enhances anti-tumor efficacy of PD-L1 blockade in syngeneic murine tumor models.

Points to be considered:

1. When discussing the methodology applied for number selection it is not clear for my understanding, how the author selected the sufficient experimental setting. Indeed, in the in vivo experiments, sample size has been should be calculated in a rigorous way, i.e. by using G*Power software (power of for example 80% and 0.05 statistical level, etc.). Assuming an effect-size of for example, 0.4 with statistical significance of α <;0.05 and a power of 80%. A given 9 mice for each group for a total of 18 mice were extimated. This number should be increased to 20 considering an expected drop-out rate of 10% for the treatment. Can the authors comment on this?

2. Did the authors normalize for time exposure immunofluorescence in figure 1 and suppl. 1?

3. Did the authors employ unstained/isotype control compensation in fig. 4?

4. Did the author checked for normally distributed (Gaussian) data before performing parametric tests?

5. Available research, patent analysis and existing platform for the cross-talk between angiogenetic addicted and potentially immunosensitive cancers are already available can be ameliorated: describe how the authors' manuscript goes beyond the state-of-the-art, and the extent to the proposed work is ambitious. This reviewer personally misses some state-of-the-art standpoint regarding angiogenesis and immune-infiltration in cancer. As is now well known, tumors grow and evolve through a constant crosstalk with the surrounding microenvironment, and emerging evidence indicates that angiogenesis and immunosuppression frequently occur simultaneously in response to this crosstalk. Accordingly, strategies combining anti-angiogenic therapy and immunotherapy seem to have the potential to tip the balance of the tumor microenvironment and improve treatment response (refer to PMID: 32456352 and expand introduction/discussion).

6.Innovation: describe where the proposed work is positioned in terms of research and innovation R&I maturity (from the idea of application to lab to market/clinic), providing an indication of the Technology Readiness Level, possibly distinguishing the start and by the end of the manuscript.

7. Methodology: the authors should be able to describe and explain the overall methodology, including CONCEPT, MODELS and ASSUMPTIONS that underpin this project. It should be explained how this will enable to deliver the project’s objectives. Any important challenges of the chosen methodology should be identified and methods to overcome them should be clearly stated.

8. Few bullets regarding a computationally specific approach are also needed, referring to the EFFECTS of the projects, and R&I in general in this field technological outcomes: the manuscript should bring new products, services, processes to the clinician, increasing efficiency (quantify), decreasing costs, increasing profits, decreasing mortality, and improving decision making in the oncology field.

Remarkably, the manuscript should give an indication of the SCALE and significance of the project’s contribution to the expected outcomes and impacts, should the project be successful (High risk/high gain balance).

SCALE refers to how the outcomes and impacts are likely to be, i.e. in terms of the size of the target group or the proportion of that group, that should benefit over time; SIGNIFICANCE refers to the importance or value of those benefits (number of additional healthy life years, etc.)

It should be always explained at the baseline, benchmarks and assumptions used for those estimates. Wherever possible, quantify the estimation of the effects expected from the project. Only one methodology should be used for calculating the estimates for each region and country.

6. PLOS authors have the option to publish the peer review history of their article (what does this mean?). If published, this will include your full peer review and any attached files.

Reviewer #1: **Yes: **Alessandro Poggi MSc, MD

Reviewer #2: No

---

## [Author Response · Author response to Decision Letter 0]

3 Apr 2022

Francesco Bertolini, MD, PhD

Academic Editor

PLOS ONE

Dear Dr. Bertolini, 

On behalf of myself and my co-authors, I would like to thank the reviewers for their valuable feedback and for the time they devoted to reviewing the article. We have addressed all feedback with a point-to-point response below (the reviewers’ comments are reproduced in italic and authors responses are in blue).

Tracked and clean versions of our resubmission addressing the points raised have been provided.

We hope that our letter and corresponding changes in the manuscript in response to the reviewers' comments are satisfactory.

Reviewers' comments:

Reviewer #1: This manuscript deals with the effect on tumor growth of a combination of anti-VEGFR2 and anti-PDL1 antibodies in murine models.

It is evident that the combination therapy is more effective compared to monotherapy. This effect is accompanied by a stronger T cell infiltration, activation of macrophages and immunological memory to further rechallenge with tumor cells.

Overall, the manuscript is well organized and written and the flow of experiments are straightforward.

I suggest some improvements to further reinforce the main message of this work.

a- the analysis of infiltrating lymphocytes is not well described. Indeed, it is not clear how large are the tumors analyzed, whether a whole scan of tumor sections have been performed, and whether a software of analysis has been used to get the level of infiltration. Also, no analysis of markers involved in lymphocyte migration and localization has been considered. It is conceivable that T cells and myeloid cells can home to the tumor, expressing some receptors involved in endothelial migration.

We thank the reviewer and apologize for the omission of important details on how IF and IHC were analyzed. We have now added them to the methods section (lines #219-239). We were limited in the number of markers explored, and unfortunately, we did not use lymphocyte migration markers in our analysis. The reviewer is correct, specific migration markers would have been very informative in determining accurately the amount of T cell infiltration.

b- it is not stated that the cell lines used in the tumor murine models are indeed PDL-1 positive, this is important to support the treatment with anti PDL-1 antibodies.

Thank you for this request. We have now added the appropriate references showing that the cell lines used in the mouse models are indeed PD-L1 positive (line #205).

c- all the data are coming from the in vivo murine models. It would be fine to demonstrate that TIL, splenic cells, lymph node T cells, PB lymphocytes can recognize the tumor cells in vitro. In other words, are CTLs present in the mice-treated specific for the tumor cells used to induce tumors? Do NK cells play a role in killing tumor cells, and can the antibodies used for therapy trigger ADCC?

It’s a great question, and unfortunately, we don’t know the answer. Due to limitation in tumor size, it would have been challenging to determine whether tumor TILs in treated animals responded specifically to tumors. We could have used peripheral blood or splenic T cells, but unfortunately, we didn’t do it, and there is no guarantee these cells represent the same T cells present in tumors. We included NK markers in flow analysis, but we did not detect a change of percentages or absolute numbers of NK cells in the TME after treatment, suggesting NK cells are not critical in the responses observed. Finally, we know that DC101 can’t trigger ADCC, but PD-L1 Ab 178G7 has not been tested for ADCC activity, so we cannot fully answer the reviewer’s question. We have added all these caveats to the discussion (lines #438-444).

“This suggests that the increased T cell infiltration observed after DC101 therapy resulted not only from inhibition of tumor angiogenesis but also through activation of the vasculature causing T cell recruitment. It is also possible that tumor infiltrated T cells are further activated by the treatment to recognize tumor antigens, but our experiments were not designed to address this question and further research will be needed. Finally, we did not observe any changes in NK cell percentages or counts, suggesting NK cells are not critical driving the observed response.”

Reviewer #2: Li et al. uncovered the treatment with a VEGFR-2 antibody to be potentially associated with intra-tumor immune modulation and enhances anti-tumor efficacy of PD-L1 blockade in syngeneic murine tumor models.

Points to be considered:

1. When discussing the methodology applied for number selection it is not clear for my understanding, how the author selected the sufficient experimental setting. Indeed, in the in vivo experiments, sample size has been should be calculated in a rigorous way, i.e. by using G*Power software (power of for example 80% and 0.05 statistical level, etc.). Assuming an effect-size of for example, 0.4 with statistical significance of α <;0.05 and a power of 80%. A given 9 mice for each group for a total of 18 mice were extimated. This number should be increased to 20 considering an expected drop-out rate of 10% for the treatment. Can the authors comment on this?

The reviewer is correct: power calculations indicate that a larger number of animals is generally needed to unequivocally determine small effect sizes like the ones observed in this study. However, there are also practical reasons that are critical when planning animal studies, and it’s not always straightforward to design studies that account for both, statistical and practical factors: animal welfare, cost, animal housing space available, and the ability to take down animals and to handle samples sequentially in a given period of time, while preserving tumor factors that easily degrade if not processed in a timely fashion. 5-6 animals per treatment arm is a standard number used in animal studies that allows a balance between detecting signals, cost and effectively handling tumor samples. Further, 5-6 animals per treatment arm is widely used in the literature.

2. Did the authors normalize for time exposure immunofluorescence in figure 1 and suppl. 1?

Thank you for this question. We failed to indicate that isotype controls were used in all experiments to ensure immunofluorescence signals were properly normalized. This has been now clarified in the methods section (line #230). “with appropriate isotype control IgG antibodies”

3. Did the authors employ unstained/isotype control compensation in fig. 4?

Thank you for the request. We did not use isotype controls, but we used unstained and “Fluorescence Minus One” controls. This has now been clarified in the methods section (line #248)

“Appropriate unstained and “Fluorescence Minus One” controls were used.”

4. Did the author checked for normally distributed (Gaussian) data before performing parametric tests?

Yes, this was confirmed in each analysis. Model diagnostics were reviewed to confirm that parametric statistical analysis assumptions (i.e. normally distributed error with constant variance) were sufficiently met for all reported parametric analysis results. 

5. Available research, patent analysis and existing platform for the cross-talk between angiogenetic addicted and potentially immunosensitive cancers are already available can be ameliorated: describe how the authors' manuscript goes beyond the state-of-the-art, and the extent to the proposed work is ambitious. This reviewer personally misses some state-of-the-art standpoint regarding angiogenesis and immune-infiltration in cancer. As is now well known, tumors grow and evolve through a constant crosstalk with the surrounding microenvironment, and emerging evidence indicates that angiogenesis and immunosuppression frequently occur simultaneously in response to this crosstalk. Accordingly, strategies combining anti-angiogenic therapy and immunotherapy seem to have the potential to tip the balance of the tumor microenvironment and improve treatment response (refer to PMID: 32456352 and expand introduction/discussion).

Thank you very much for this suggestion. We agree with the reviewer and have now added a paragraph in the introduction to expand the crosstalk between angiogenesis and aberrant tumor responses (lines #120-127)

“Angiogenesis, the formation of new blood vessels from pre-existing ones, is an important step in cancer progression [1]. Tumor-induced angiogenesis is highly dysregulated compared to normal angiogenesis, and results in structurally aberrant vessels with insufficient pericyte coverage and leaky nascent vessels [2]. The interplay between aberrant tumor vessels and protumoral immune cells generates a vicious cycle that severely disturbs anti-cancer responses and promotes tumor progression; abnormal tumor vessels promote immune cell evasion, which in turn enhances tumor angiogenesis. This aberrant crosstalk hinders T-cell infiltration into the tumor and impairs T-cell effector functions [3-6].”

6.Innovation: describe where the proposed work is positioned in terms of research and innovation R&I maturity (from the idea of application to lab to market/clinic), providing an indication of the Technology Readiness Level, possibly distinguishing the start and by the end of the manuscript.

Prior work referenced in our manuscript showed that combination of checkpoint inhibitors and anti-angiogenic treatments leads to enhanced antitumor activity, but there was limited evidence of immunological memory. Our results shows that in addition to enhanced antitumor activity, DC101+anti-PD-L1 treatment also leads to immunological memory. We have highlighted this in the discussion (lines #464 and #485)

“Notably, the combined treatment led to immunological memory, since animals that responded to treatment were resistant to tumor rechallenge”.

“resulting in enhanced antitumor activity and immunological memory.”

7. Methodology: the authors should be able to describe and explain the overall methodology, including CONCEPT, MODELS and ASSUMPTIONS that underpin this project. It should be explained how this will enable to deliver the project’s objectives. Any important challenges of the chosen methodology should be identified and methods to overcome them should be clearly stated.

We apologize, but we do not understand what exactly the reviewer is asking us to do. We believe that with the new modifications, the methods section is now clear and concise. We are happy to incorporate any further suggestions the reviewer may have.

8. Few bullets regarding a computationally specific approach are also needed, referring to the EFFECTS of the projects, and R&I in general in this field technological outcomes: the manuscript should bring new products, services, processes to the clinician, increasing efficiency (quantify), decreasing costs, increasing profits, decreasing mortality, and improving decision making in the oncology field.

We agree with the reviewer that it is important to highlight how this work impacts the field. We believe that the current discussion (lines #478-482) highlights the importance of this preclinical work in the field of angiogenesis and checkpoint inhibition, and the impact in clinical development of ramucirumab. Perhaps the reviewer can suggest how we could do this better?

“Several clinical trials combining the anti-VEGFR-2 antibody, ramucirumab, and PD1 or PD-L1 antibodies are ongoing in several indications, including non-small-cell lung cancer, recurrent/metastatic head and neck squamous cell carcinoma, and mesothelioma (NCT03904108, NCT03650764, NCT03502746), and early indications suggest the results of this combination will be promising.

Remarkably, the manuscript should give an indication of the SCALE and significance of the project’s contribution to the expected outcomes and impacts, should the project be successful (High risk/high gain balance).

SCALE refers to how the outcomes and impacts are likely to be, i.e. in terms of the size of the target group or the proportion of that group, that should benefit over time; SIGNIFICANCE refers to the importance or value of those benefits (number of additional healthy life years, etc.)

It should be always explained at the baseline, benchmarks and assumptions used for those estimates. Wherever possible, quantify the estimation of the effects expected from the project. Only one methodology should be used for calculating the estimates for each region and country.

Please see our prior responses. We believe that the current discussion section reflects well the reviewer’s concerns. It highlights the importance of this preclinical work in the field of angiogenesis and checkpoint inhibition, and the impact in clinical development of ramucirumab. Again, we are happy to incorporate any further suggestions the reviewer may have.

6. PLOS authors have the option to publish the peer review history of their article (what does this mean?). If published, this will include your full peer review and any attached files.

We thanks the journal for providing us with this option. We think it’s an excellent practice making the entire process transparent to the reader.

We thank again both reviewers for helping us improve this manuscript. Their time and contributions have been critical to improve the final manuscript.

Regards

The authors

---

## [Decision Letter · Decision Letter 1]

26 Apr 2022

Treatment with a VEGFR-2 antibody results in intra-tumor immune modulation and enhances anti-tumor efficacy of PD-L1 blockade in syngeneic murine tumor models

PONE-D-21-39856R1

Dear Dr. Puig,

We’re pleased to inform you that your manuscript has been judged scientifically suitable for publication and will be formally accepted for publication once it meets all outstanding technical requirements.

Kind regards,

Francesco Bertolini, MD, PhD

Academic Editor

PLOS ONE

Additional Editor Comments (optional):

Reviewers' comments:

Reviewer's Responses to Questions

**Comments to the Author**

1. If the authors have adequately addressed your comments raised in a previous round of review and you feel that this manuscript is now acceptable for publication, you may indicate that here to bypass the “Comments to the Author” section, enter your conflict of interest statement in the “Confidential to Editor” section, and submit your "Accept" recommendation.

Reviewer #1: All comments have been addressed

Reviewer #2: All comments have been addressed

2. Is the manuscript technically sound, and do the data support the conclusions?

Reviewer #1: Yes

Reviewer #2: Yes

3. Has the statistical analysis been performed appropriately and rigorously? 

Reviewer #1: Yes

Reviewer #2: Yes

4. Have the authors made all data underlying the findings in their manuscript fully available?

Reviewer #1: Yes

Reviewer #2: Yes

5. Is the manuscript presented in an intelligible fashion and written in standard English?

Reviewer #1: Yes

Reviewer #2: Yes

6. Review Comments to the Author

Reviewer #1: The authors have replied to all my queries. So, I endorse the manuscript for publication in PlosOne.

Reviewer #2: The authors have clarified several of the questions I raised in my previous review. Most of the major problems have been addressed by this revision.

7. PLOS authors have the option to publish the peer review history of their article (what does this mean?). If published, this will include your full peer review and any attached files.

Reviewer #1: **Yes: **Alessandro Poggi MSc, MD

Reviewer #2: No

---

## [Editor Report · Acceptance letter]

8 Jul 2022

PONE-D-21-39856R1 

Treatment with a VEGFR-2 antibody results in intra-tumor immune modulation and enhances anti-tumor efficacy of PD-L1 blockade in syngeneic murine tumor models 

Dear Dr. Puig:

I'm pleased to inform you that your manuscript has been deemed suitable for publication in PLOS ONE. Congratulations! Your manuscript is now with our production department. 

Kind regards, 

on behalf of

Dr. Francesco Bertolini 

Academic Editor

PLOS ONE